# Neuro-Symbolic Data Generation for Math Reasoning

**Zenan Li**[1*]   **Zhi Zhou**[1*]   **Yuan Yao**[1]   **Yu-Feng Li**[1]
**Chun Cao**[1]   **Fan Yang**[2]   **Xian Zhang**[2]   **Xiaoxing Ma**[1]
[1]State Key Lab of Novel Software Technology, Nanjing University, China
[2]Microsoft Research Asia
lizn@smail.nju.edu.cn, zhouz@lamda.nju.edu.cn,
zhxian@microsoft.com, xxm@nju.edu.cn

## Abstract

A critical question about Large Language Models (LLMs) is whether their apparent deficiency in mathematical reasoning is inherent, or merely a result of insufficient exposure to high-quality mathematical data. To explore this, we developed an automated method for generating high-quality, supervised mathematical datasets. The method carefully mutates existing math problems, ensuring both diversity and validity of the newly generated problems. This is achieved by a neuro-symbolic data generation framework combining the intuitive informalization strengths of LLMs, and the precise symbolic reasoning of math solvers along with projected Markov chain Monte Carlo sampling in the highly-irregular symbolic space. Empirical experiments demonstrate the high quality of data generated by the proposed method, and that the LLMs, specifically LLaMA-2 and Mistral, when realigned with the generated data, surpass their state-of-the-art counterparts.

## 1   Introduction

Despite recent progress [1–6], both proprietary and open-source LLMs are still far from satisfactory in mathematical reasoning [7–9]. It is an open question whether LLM's subpar reasoning capability is inherent or due to the the extreme scarcity of high-quality mathematical datasets [10–13]. As an initial step towards answer this question, a data generation framework that could create high-quality math datasets is required. To this end, current two lines of research struggle in the *diversity-validity* dilemma: (1) to produce diverse math data, the prompt-based method effectively rephrases math problems using LLMs, but may induce errors thus ruining the *validity*, especially considering the rigor of maths; (2) to ensure the validity, template-based methods are often used by rewriting math problems with certain rules, sacrificing the *diversity* and thus confining data scale.

To address this dilemma, we propose a novel *neuro-symbolic* framework that automatically generates high-quality, supervised mathematical data. The merit of this paradigm lies in leveraging both neural and symbolic strengths: (1) the math problem is generated in the symbolic space, achieving diversity through systematic sampling, while maintaining validity through symbolic solvers; (2) the translation from the symbolic space back to the natural language space can be effectively supported by LLMs, ensuring the consistency between newly generated formal problems and their corresponding natural language versions.

Our framework, as illustrated in Figure 1, initiates with the formalization of the original problem expressed via the math symbolic tools. Next, it *mutates* the formal problem into an evolved version, and then derives a new natural language problem by *informalization*. Specifically, we design a mutation mechanism, including various simplification and complication strategies, such that the new problems can be generated with a controllable complexity. As shown in Figure 2, our mutation

---

[*]Equal contribution. This work was partially done during Zenan's internship at MSRA.

38th Conference on Neural Information Processing Systems (NeurIPS 2024).

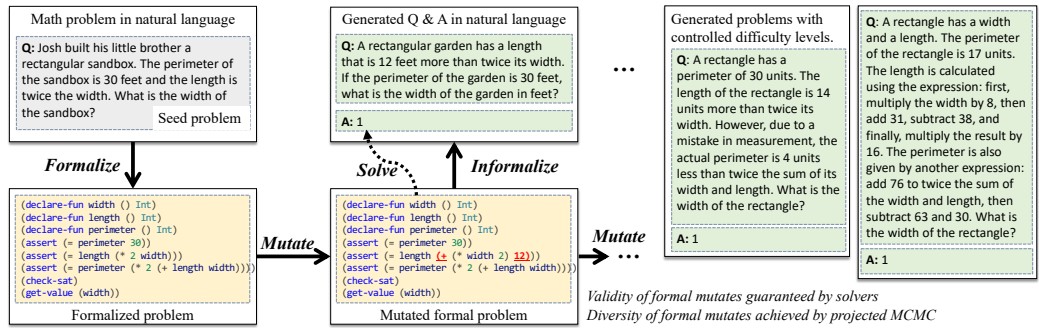

Figure 1: The overview of our neuro-symbolic data generation framework. The framework comprises three steps: (1) Formalize the seed problem into its symbolic version. (2) Mutate the symbolic problem to create new variants. (3) Translate the variants in symbolic form back to the natural language version. Additionally, we prompt GPT-4 to generate reasoning paths, which are verified by symbolic solvers, as part of the supervision.

mechanism can properly adjust the complexity of generated problems, and the exposure to more complex math problems can improve the LLM's reasoning capability. Moreover, to ensure the data validity and achieve higher generation diversity, we combine the symbolic solving with the random sampling through the projected Markov chain Monte Carlo technique [14, 15].

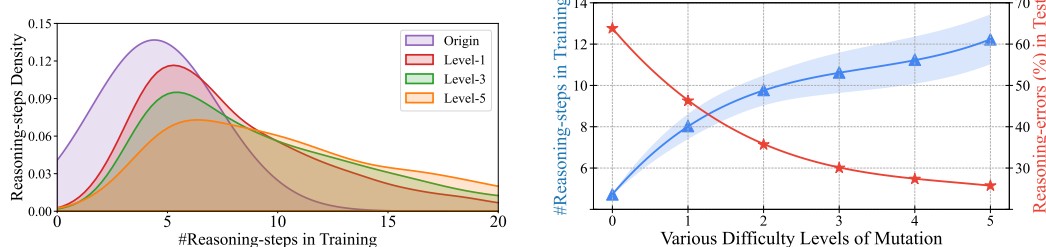

Figure 2: The performance of our proposed mutation mechanism. The first figure illustrates that the generated problems with higher difficulty levels lead to more reasoning steps of GPT-4. The second figure shows that the gradual incorporation of more difficult problems consistently improves the LLM's reasoning capability.

Empirical evaluation on GSM8K [10] and MATH [11] demonstrates the effectiveness of the proposed method. Particularly, we use the proposed framework to generate a mathematical dataset of 620K examples for supervised fine-tuning. The experimental results show that, the fine-tuned models on LLaMA-2 [16] and Mistral-7B [17] significantly outperform the existing open-source counterparts on both GSM8K and MATH datasets, as well as two out-of-domain datasets SVAMP [12] and ASDiv [18]. On the GSM8K dataset, the model fine-tuned on Mistral-7B even outperforms GPT-3.5-Turbo (by 2.4%), a proprietary model with an order of magnitude larger parameters. Additionally, we evaluate the scalability of our method, and observe consistent performance improvements, as the size of training data increases. This upward trajectory suggests a promising avenue for further enhancing LLMs' mathematical capabilities.

## 2   Mutation

Compared to existing data generation methods, the key feature of our framework lies in the mutation of math problems within the symbolic space, enabling systematic sampling and symbolic solving. Technically, our mutation mechanism includes several simplification and complication strategies, to control the complexity of the generated math problems. The overall framework of our problem mutation method is summarized in Algorithm 1.

---

**Algorithm 1** The overall framework of problem mutation

---

**Input:** A seed problem expressed by goal $g$ and constraints $h_1, \ldots, h_n$.
**Output:** A new problem expressed by goal $g'$ and constraint $h'_1, \ldots, h'_n$.

1: **for** $i = 1, \ldots,$ **do**                     ▷ *Complicate the problem using Projected MCMC*
2:      Initialize random operations $\oplus_i$ and interpreted functions $e'_i$.
3:      Mutate $g' = g \oplus_0 e'(z_0)$, $h'_i = h_i \oplus_i e'(z_i)$ with parameter $z_i$.
4:      Randomly perturb a subset $\{z_1, \ldots, z_j\}$ for $j < n$.
5:      Solve the rest subset $\{z_{j+1}, \ldots, z_n\}$.
6:      **if** $\{z_1, \ldots, z_n\}$ is solvable **then**
7:          Instantiate the new problem by $\{z_1, \ldots, z_n\}$.
8:          Stop the complication process
9:      **end if**
10: **end for**
11: **for** $i = 1, \ldots$ **do**                      ▷ *Simplicate the problem using SMT tactics*
12:      Initalize a random tactic from $\{simplify, qe, simplify, \ldots\}$.
13:      Apply the tactic to the problem $g'$ and constraints $h'_1, \ldots, h'_n$.
14: **end for**

---

## 2.1 Formalization

We first provide the formalization of math problems, based on which the mutation mechanism is operated. Specifically, we adopt the SMT-LIB language [19], a standard notation compatible with prevalent SMT solvers (e.g., Z3 [20], CVC5 [21], and MathSAT [22]). It can also be easily extended for symbolic calculators (e.g., SymPy [23]) and numerical solvers (e.g., SciPy [24]). With SMT-LIB language, the math problem in the following structure is enabled:

$$
\begin{aligned}
\textit{Goal} \qquad & g := \min \mid \max \mid \text{solve } f(\mathbf{x}) \\
\textit{Constraints} \quad & h := h_1 \wedge h_2 \mid h_1 \vee h_2 \mid \text{ite}(h_1, h_2, h_3) \mid \\
& \qquad \forall \mathbf{x}.\, e_1(\mathbf{x}) \bowtie e_2(\mathbf{x}) \mid \exists \mathbf{x}.\, e_1(\mathbf{x}) \bowtie e_2(\mathbf{x}) \mid \\
& \qquad e_1 \bowtie e_2, \quad \bowtie \in \{\geq, \leq, >, <, =, \neq\} \\
\textit{Expressions} \quad & e := c \mid \mathbf{x} := (x_1, \ldots, x_n) \mid \text{foo}(\mathbf{x}) \mid \\
& \qquad e_1 \oplus e_2, \quad \oplus \in \{+, -, \times, \div\} \\
\textit{Domains} \qquad & \mathcal{D} := \mathbb{N} \mid \mathbb{N}^+ \mid \mathbb{R} \mid \mathbb{C}
\end{aligned}
$$

where $c$ denotes a constant, $\mathbf{x}$ denotes an *n*-dimensional variable, ite denotes the if-then-else structure, foo refers to an *interpreted* function (e.g., trigonometric, logarithmic or user-defined ones) on the domain, and $g$ and $h$ represent any function of interest (can include quantifiers). In particular, we pre-defined a series of interpreted functions, such as summation, binomial, gcd, lcm, derivate, and integral, which facilitate the formalization of most high-school level mathematical problems (excluding geometry) within the above SMT-LIB language.

## 2.2 Simplification

We perform simplification by systematically considering *expression reduction* and *constraint reduction*, which can be attained through heuristic tactics provided by standard symbolic solvers [25].

Specifically, we apply the *simplify* tactic for expression reduction, which involes operations such as constant or variable folding (e.g., $x + 0 \Rightarrow x$ or $y + x - x \Rightarrow y$), expression expansion (e.g., $(x+1)^2 \Rightarrow x^2 + 2x + 1$), and function application (e.g., $(x = 2) \wedge (y = \log(x)) \Rightarrow y = \log(2)$); we also perform symbolic and numerical computations for further reductions (e.g., $\gcd(2x, 6y) \Rightarrow 2\gcd(x, 3y)$ and $\sin(\pi/6) \Rightarrow 0.5$).

For constraint reduction, we mainly employ the Gaussian elimination tactic *gaussian_elim* (e.g., $x = 2 \wedge y \leq x + z \Rightarrow y \leq 2 + z$). To handle the if-then-else term, we apply the *elim_term_ite* tactic to decompose it by introducing a fresh variable (e.g., $\text{ite}(x > y, x, y) > z \Rightarrow (k > z) \wedge (x > y \rightarrow k = x) \wedge (x \leq y \rightarrow k = y)$). For constraints involving quantifiers, we strive to eliminate them using the *qe* tactic (e.g., $\exists y.(y > 0) \wedge (x = y + 2) \Rightarrow x > 2$). Appendix B provides more examples illustrating these simplifications.

## 2.3 Complication

To *complicate the expressions*, a straightforward strategy is to incorporate additional operators. For example, given an atomic constraint $h = e_1 \bowtie e_2$, we can introduce an additional expression, denoted by $e'$, and derive a more complex constraint $\tilde{h} = e_1 \bowtie (e_2 \oplus e')$.

$$(M_1) \begin{cases} a(b+c) = 152 \\ b(c+a) = 162 \\ c(a+b) = 170 \\ a,b,c \in \mathbb{N}^+ \end{cases} \Rightarrow (M_2) \begin{cases} a(b+c) = 152 \oplus_1 e'_1(z_1) \\ b(c+a) = 162 \oplus_2 e'_2(z_2) \\ c(a+b) = 170 \oplus_3 e'_3(z_3) \\ a,b,c \in \mathbb{N}^+, z_1, z_2, z_3 \in \mathbb{R} \end{cases}$$

$$\Rightarrow (M_3) \begin{cases} a(b+c) + z_1 = 152 \\ b(c+a) - z_2 = 162 \\ c(a+b) = 170 \\ z_1 = 114 \\ z_2 = 36 \\ a,b,c \in \mathbb{N}^+ \end{cases} \Rightarrow (M_4) \begin{cases} a(b+c) + d = 152 \\ b(c+a) - e = 162 \\ c(a+b) = 170 \\ d + e = 150 \\ d - e = 78 \\ a,b,c,d,e \in \mathbb{N}^+ \end{cases}$$

However, such a strategy is non-trivial in practice. The first challenge lies in the *validity* aspect, i.e., the math problem is often carefully designed, and thus a random mutation may ruin their well-defined structure. Consider the running example problem $(M_1)$, which has been normalized for simplicity. In this problem, a reckless mutation can easily violate the positive integer constraints, causing the problem ill-defined and unsolvable.

To address this issue, we equip each mutation with an auxiliary variable, followed by symbolic solvers to ensure the problem remains well-defined. Continuing with the previous example, we introduce three auxiliary variables, denoted by $z_1, z_2, z_3$, and then mutate the problem as $(M_2)$, where $\oplus_1, \oplus_2, \oplus_3 \in \{+, -, \times, \div\}$ represent three random operators. Furthermore, we instantiate $e'_1, e'_2, e'_3$ by interpreted functions, i.e., $e'_1 = \text{foo}_1(z_1)$, $e'_2 = \text{foo}_2(z_2)$, and $e'_3 = \text{foo}_3(z_3)$, where $\text{foo}_1, \text{foo}_2, \text{foo}_3$ are randomly selected from $\text{foo}(z) = z \mid \log(z) \mid \exp(z) \mid \arcsin(z) \mid \cdots$. For our running example, we simply choose the identity function for $\text{foo}_1, \text{foo}_2, \text{foo}_3$, and set $\oplus_1 = -, \oplus_2 = +, \oplus_3 = \times$. Using symbolic solvers to compute a feasible solution of $(z_1, z_2, z_3)$, we derive a new and well-defined problem $(M_3)$.

The subsequent challenge is to ensure the *diversity* of the mutated problems, which now becomes how to make the solutions of auxiliary variables sufficiently diverse. This is essentially a model counting problem [26, 27], and current symbolic solvers still underperform in this regard [28]. To this end, we instead opt for auxiliary variable solution generation via the *projected Markov chain Monte Carlo* (projected MCMC) [14, 15]. Simply put, projected MCMC first perturbs a subset of variables (projected random walk), and then resolves the remaining part (inverse projection via symbolic solvers), which ensures both diversity and validity of the variable solutions.

Finally, to *complicate the constraints*, one can easily reverse the process of simplification. For our running example, we can reverse the Gaussian elimination with refreshed variables, obtaining the final form $(M_4)$, which is then included as a new problem in the dataset.

## 3 Informalization

Informalization aims to translate a formal problem back to natural language without the loss of soundness [29]. As shown in Example 1, a simple, one-line instruction follows the formally posed SMT-LIB problem, serving as the input. Then, GPT-4 interprets the formal problem as a new math word problem.

The key challenge of informalization lies in ensuring a *consistent* conversion, i.e., the natural language problem informalized by GPT-4 should align with the formal solution given by symbolic solvers. Since it is difficult to directly measure this consistency, we instead use GPT-4 to generate a solution for each informalized problem, and then calculate the *consistency rate* between the solutions from GPT-4 and those from symbolic solvers as a surrogate metric. Furthermore, we observe that, if the problem is *incorrectly* informalized, GPT-4's solutions almost cannot be confirmed by symbolic solvers (i.e., zero false positive). Therefore, the surrogate consistency rate can be regarded as a lower bound to the true consistency rate.

**Example 1: Informalization**

```
(declare-fun sara_shoes_cost () Real)
(declare-fun sara_dress_cost () Real)
(declare-fun sara_total_cost () Real)
(declare-fun rachel_budget () Real)
(assert (= sara_shoes_cost 50.0))
(assert (= sara_dress_cost 200.0))
(assert (= sara_total_cost (+ sara_shoes_cost sara_dress_cost)))
(assert (= rachel_budget (* 2 sara_total_cost)))
(check-sat)
(get-value (rachel_budget))
```

Translate the math problem formulated with SMT-LIB back to a natural language problem.

- - - - - - - - - - - - - - - - - - - - - - - - - - - - - - - - - - - - - - - - - -

**GPT-4 output:**
Sara bought a pair of shoes for $50.00 and a dress for $200.00. If Rachel has twice the amount that Sara spent in total, how much is Rachel's budget?

To further improve conversion consistency, in addition to the basic zero-shot learning template in Example 1, we investigate the effects of the following operations, whose detailed examples are available in Appendix C.

(1) **Mutation**. Mutation complicates the problem, making the informalization more difficult. Therefore, we first analyze the informalization error caused by the mutation.

(2) **Few-shot learning**. Few-shot examples offer a stronger instruction to the LLM, and also introduce the randomness when aided by random retrieval.

(3) **Comment generation**. Recognizing that GPT-4 is unfamiliar with SMT-LIB's prefix expressions, we automatically convert these into the infix format, included as comments.

(4) **Math-word instruction**. We simply append one more sentence "Ensure to be a math word problem" in the prompt. With this instruction, informalization tends to imbue digits with some practical meaning (e.g., $7 \Rightarrow$ one week).

Table 1: Consistency rate of six different operations used in informalization: (1) Mutation; (2) Few-shot learning; (3) Comment generation; (4) Math-word instruction; (5) Problem modification; (6) Variable refresh. We recommend two patterns (i.e., P1: 1-5 and P2: 1-3&6), both of which can achieve satisfactory results.

| Ops | (1) | (2) | (3) | (4) | (5) | (6) | Rate (%) |
|-----|-----|-----|-----|-----|-----|-----|----------|
| –   | ✗   | ✗   | ✗   | ✗   | ✗   | ✗   | 75.6 (–) |
|     | ✓   | ✗   | ✗   | ✗   | ✗   | ✗   | 41.6 (↓) |
| –   | ✓   | ✓   | ✗   | ✗   | ✗   | ✗   | 76.2 (↑) |
|     | ✓   | ✓   | ✓   | ✗   | ✗   | ✗   | 87.6 (↑) |
| P1  | ✓   | ✓   | ✓   | ✓   | ✓   | ✗   | 90.5 (↑) |
| P2  | ✓   | ✓   | ✓   | ✗   | ✗   | ✓   | 97.1 (↑) |

(5) **Problem modification**. For mutated problems, rather than generating a new informalization, we prompt GPT-4 to modify the original informalization result.

(6) **Variable refresh**. We standardize the naming of all introduced variables (e.g., `rachel_budget` $\Rightarrow$ `x_1`), to eliminate the impact of math word problems.

Different combinations of the above operations result in different patterns. The effects of some typical patterns are shown in Table 1, where the results are evaluated on 1,000 problems randomly sampled from GSM8K. The basic pattern in Example 1 yields a consistency rate of 75.6%. The mutation operation alone indeed degrades the consistency, but its combination with other operators can further boost the informalization performance. In practice, we use two different patterns for different informalization styles: the first pattern (P1) tends to generate math word problems, whereas the problems generated by the second pattern (P2) tend to be pure math problems.

# 4 Experiments

In this section, we conduct a series of experiments to answer the following four research questions:

**RQ1: Efficacy** – Using our data generation framework, can the fine-tuned model achieve better performance compared with existing models?

**RQ2: Efficiency** – Given the same data generation budget, is the generated data from our framework better than that from the state-of-the-art data generation framework?

**RQ3: Generability** – Is the effecitveness achieved by our framework due to potential data contamination introduced during the generation process?

**RQ4: Scalability** – With more data generated, can our approach be continually effective in further improving model performance?

## 4.1 Experimental Setup

**Dataset**. We conduct our data generation on the training sets of two popular mathematical reasoning benchmarks: GSM8K [10] and MATH [11]. GSM8K is a dataset comprising high-quality grade school math problems, which contains 7,473 training data and 1,319 testing data. MATH is a dataset comprised of challenging competition math problems, spanning seven subjects including Prealgebra, Algebra, Number Theory, Counting and Probability, Geometry, Intermediate Algebra, and Precalculus. There are 7,500 training data and 5,000 testing data in the MATH dataset. Additionally, we include two mathematical reasoning datasets, i.e., SVAMP [12] and ASDiv [18], to evaluate the out-of-domain generalizability of the models fine-tuned on the data generated from GSM8K and MATH datasets.

**Comparison Methods**. In our experiments, we compare the models trained using our generated data with existing state-of-the-art open-source mathematical reasoning models, including WizardMath [30], MuggleMATH [31], MAmmoTH [32], and MetaMath [33]. We also conduct a thorough comparison between our math generation method and the bootstrapping method employed in MetaMathQA [33], which is presently the most extensive open-source dataset for mathematical reasoning.

**Data Generation Details**. We use our mutation mechanism to generate a series of problems with varying levels of difficulty, and the specifics are as follows. Starting with a problem from the original dataset as a seed, we first perform simplification to the problem, and define this new version as level-0. Then, we randomly apply one expression complication step and one constraint complication step to the level-0 version, deriving a more difficult problem (level-1 version); and such complications can be repeated to obtain more difficult problems. For the GSM8K dataset, we create datasets across five levels of difficulty, with 30K examples at level-0 and 100K examples for the remaining four levels. As for the MATH dataset, we establish four levels of difficulty, where level-0, level-1, level-2, and level-3 contain 70K, 120K, 120K, and 120K examples, respectively. Particularly, for some problems in the MATH dataset that cannot be solved by symbolic solvers, we directly prompt GPT-4 to rephrase the problem and ignore the solution verification. The number of generated problems without solution verification varies across problem categories, and the details can be referred to Appendix D. In total, we generated 860K math problems based on the proposed framework to construct our dataset.

Each generated math problem consists of a natural language problem description informalized by GPT-4 (version 0710), a final answer outputted by the symbolic solver, and a reasoning path from the problem to the final answer. The reasoning path for each problem is also generated by GPT-4, which is further verified by the corresponding answer derived from symbolic solvers.

More implementation details about training hyperparameters, instruction prompts, and symbolic solver integration, are included in Appendix D.

## 4.2 Empirical Results

**RQ1: Efficacy**. Using the generated math dataset, we fine-tune the LLaMA-2 base models of 7B and 13B parameter sizes, as well as the Mistral 7B base model. The fine-tuned models, as well as the comparison methods, are evaluated on the GSM8K and MATH datasets.

Table 2: Performance comparison among existing mathematical reasoning models fine-tuned on three base models (LLaMA-2 7B, LLaMA-13B, and Mistral 7B). The best performance is in bold. The delta performance between our model and other SOTA LLMs on each dataset is also reported.

| Model | #Dataset | LLaMA-2 7B Base | | LLaMA-2 13B Base | | Mistral 7B Base | |
| --- | --- | --- | --- | --- | --- | --- | --- |
| | | GSM8K | MATH | GSM8K | MATH | GSM8K | MATH |
| WizardMath | >240K | 54.9 | 10.7 | 63.9 | 14.0 | 83.2 | 33.0 |
| MuggleMATH | 157K | 68.4 | 8.4 | 74.0 | 9.4 | - | - |
| MAmmoTH† | 260K | 50.5 | 10.4 | 56.3 | 12.9 | 61.9 | 17.5 |
| MetaMath | 395K | 66.5 | 19.8 | 72.3 | 22.4 | 77.7 | 28.2 |
| Ours | 860K | **79.0** | **30.4** | **84.1** | **33.7** | **86.8** | **37.3** |
| Δ | | ↑ 10.6 | ↑ 10.6 | ↑ 10.1 | ↑ 11.3 | ↑ 3.6 | ↑ 4.3 |

† Model performance is re-evaluated using Pass@1 of CoT prompt.

As shown in Table 2, our approach achieves the best performance among the baseline models across different model scales. Compared to LLMs with the LLAMA-2 7B base model, our model demonstrates a fair improvement in accuracy, surpassing them by at least 10.6% on the two datasets. For our model fine-tuned using the LLAMA-2 13B base model, our model achieves an accuracy of 84.1% and 33.7%, outperforming existing SOTA model by 10.1% and 11.3%. When fine-tuned on the Mistral 7B base model, our model still attains the best performance with an increase in accuracy of 3.6% and 4.3%, respectively. Notably, our model even slightly outperforms GPT-3.5-Turbo (80.8% and 34.1%) by 6.0% on the GSM8K dataset and 3.2% on the MATH dataset.

In addition to the above competitors, we also compare our models with tool-based models, and provide the results in Appendix E.2. We summarize two observations here. First, tool-based models tend to over-rely on external tools, and thus do not necessarily improve the inherent reasoning ability of LLMs. Second, although the tool-based models perform better on the MATH dataset (which frequently entails complex calculations), they still underperform on the GSM8K dataset (which emphasizes knowledge reasoning but involves simpler calculations).

**RQ2: Efficiency**. To illustrate the data efficiency of our method, we carry out a comparative experiment with current SOTA method MetaMathQA [33]. The MetaMathQA dataset comprises 240K data bootstrapped from the GSM8K training dataset and 155K data from the MATH training dataset. To ensure a fair comparison with MetaMathQA, we use the same data budget. Additionally, we expand the MATH data of the MetaMathQA dataset to 430K, aligning its size with that of our generated data. Then, we fine-tune LLaMA-2-7B models on the 240K GSM8K augmented data, as well as the 155K and 430K MATH augmented data, respectively.

The performance of fine-tuned models are given in Table 3. We also evaluate the models on two out-of-domain datasets, SVAMP and ASDiv. The results confirm the efficiency of our framework. With an equal generation budget of 240K for the GSM8K dataset and 155K for the MATH dataset, our method exhibits accuracy improvements, ranging from 2.1% to 24.4% across the four datasets. This superiority is consistent with the model trained on 430K MATH generation data, with improvements of 19.9%, 3.2%, 12.7%, and 19.2%, respectively.

**RQ3: Generability**. Despite we carefully ensure that our mutations on the training set do not access the test set, we still provide a series of analysis about the potential data contamination or overfitting issues. We first use the memorization detection method introduced in Minerva [34]. Specifically, we select 150 problems with the highest majority vote score and then compute the BLUE score [35] on solutions of trained Mistral 7B, GPT-4, and the ground-truth. The results of our method and MetaMath are provided in Figure 3, which show that our BLUE score on the test set is (1) much lower than our BLUE score on the training set; (2) is consistent with that of MetaMath. Hence, there is no evidence that the mutation contaminates the test set.

Second, in addition to the two out-of-distribution datasets SVAMP and ASDiv, we conduct experiments on another benchmark DyVal [9], which avoids the leak of test set through dynamically generating new benchmarks. The results of our models, alongside those of the comparison models, are provided in Appendix E.3. In summary, our models demonstrated superior performance in 11 out

Table 3: Comparison between our method and MetaMathQA (MMQA) with the same data budgets. The models are fine-tuned using LLaMA-2-7B base model, and evaluated on GSM8K, MATH, SVAMP, and ASDiv datasets. The results illusrate the high quality of our generated data.

| Method | Training Dataset GSM8K | Math | Performance GSM8K | Math | SVAMP | ASDiv |
|---|---|---|---|---|---|---|
| MMQA | 240K | 0K | 66.1 | 5.8 | 61.7 | 72.5 |
| Ours | 240K | 0K | 72.7 | 8.2 | 78.8 | 79.2 |
| Δ | | | ↑ 6.6 | ↑ 2.3 | ↑ 17.1 | ↑ 6.7 |
| MMQA | 0K | 155K | 28.6 | 19.9 | 49.0 | 61.0 |
| Ours | 0K | 155K | 42.1 | 22.0 | 73.4 | 64.1 |
| Δ | | | ↑ 13.5 | ↑ 2.1 | ↑ 24.4 | ↑ 3.1 |
| MMQA | 240K | 155K | 67.5 | 21.7 | 64.1 | 76.0 |
| Ours | 240K | 155K | 73.7 | 23.4 | 85.2 | 81.1 |
| Δ | | | ↑ 6.2 | ↑ 1.7 | ↑ 21.1 | ↑ 5.0 |
| MMQA* | 0K | 430K | 35.0 | 25.6 | 66.4 | 51.1 |
| Ours | 0K | 430K | 54.9 | 28.8 | 79.1 | 70.3 |
| Δ | | | ↑ 19.9 | ↑ 3.2 | ↑ 12.7 | ↑ 19.2 |

Figure 3: BLUE scores between the output of our fine-tuned model, versus the ground-truth solution and GPT-4 output. The model is fine-tuned on the Mistral 7B base model, and MetaMath Mistral 7B model is also included as a reference. The results show that our method does not induce data contamination.

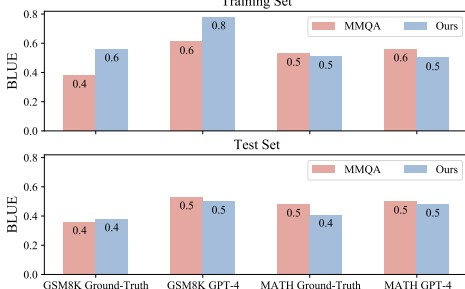

of 12 cases and delivered competitive results in the remaining case. Particularly, as the complexity of the tasks increased, our models exhibited a relatively robust performance compared to other models.

Finally, we include an additional experiment on the Hungarian High School National Finals Exam dataset [36], whose problems are newly collected at 2023. We manually check 33 testing problems based on the provided ground-truth answer, and our model correctly solved 14 problems and partially solved 6 problems, resulting in an exam score of 44. The result is comparable to GPT3.5-turbo (i.e., 41 exam score), conforming the generalizability of our method.

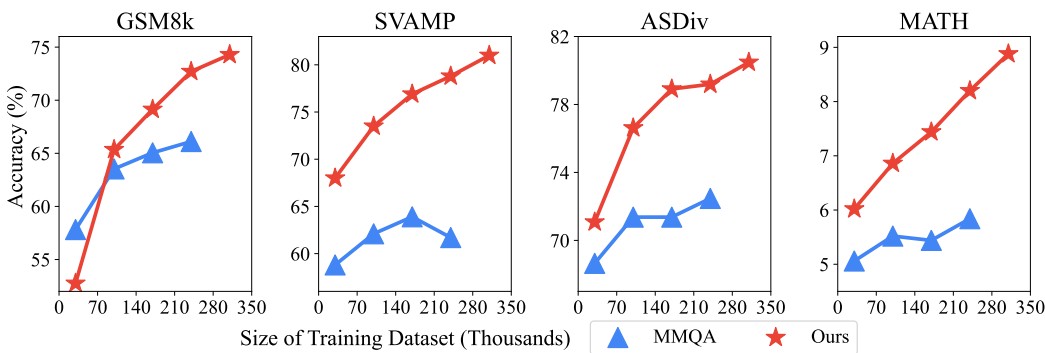

Figure 4: Performance curves of the LLaMA-2-7B models fine-tuned on various scales of datasets. The two datasets are generated by our approach and MetaMath (MMQA). The performance can be consistently enhanced by increasing the amount of data generated using the proposed framework.

**RQ4: Scability**. To explore the scalability of our framework, we fine-tune the LLaMA-2 7B model using our generated datasets of various sizes and difficulties. To be specific, we progressively incorporate a 30K dataset, along with four additional 70K datasets generated from GSM8K. Note that these five datasets are randomly sampled from five different levels of difficulty. Five LLaMA-2-7B models are fine-tuned based on these datasets, and the scalability curves are shown in Figure 4. We also include MetaMathQA with the same data settings as a reference. Since MetaMathQA cannot inherently group the dataset into various difficult levels, we construct five datasets by incrementally random sampling from the GSM8K subset of the MetaMathQA dataset.

The results presented in Figure 4 indicate the promising scalability of our method. That is, as the size of data increases, the accuracy of the model consistently improves. In contrast, the performance enhancement observed in MetaMathQA is limited and starts to diminish as the data size reaches 70K. We also present the models' performance on the other three out-of-domain datasets in this case, i.e., SVAMP, ASDiv, and MATH datasets. The results demonstrate that the scalability of our method is robust and generalizable, while MetaMathQA hardly guarantees such consistency.

We also investigate the diversity gain relative to the original dataset for each difficulty level. The results are provided in Appendix E.4. It is observed that the dataset consisting of the same difficulty level cannot further improve the diversity with a larger data budget. On the contrary, the diversity gain of the dataset comprising all difficulty levels continues to increase as the data budget grows.

## 5 Related Work

Recent surveys [37–39] have comprehensively discussed the current advances in the mathematical reasoning of LLMs. Here, we review three main lines of existing work on enhancing the mathematical reasoning for LLMs related to our study: prompt-based methods, rephrasing-based methods, and tool-based methods.

**Prompt-based Method**. Prompt-based methods aim to harness the inherent capabilities of LLMs by carefully designing appropriate input prompts without tuning the model parameters. This line of work starts from the observation that LLMs can effectively tackle more math problems when provided with a simple Chain-of-Thought (CoT) prompt, i.e., "Let's think step by step" [40]. Building upon the CoT prompt, Wang et al. [41] further propose to consolidate multiple reasoning paths based on the self-consistency of correct answers. Later, several researchers propose to prompt LLMs to decompose complex problems. For example, Zhou et al. [42] introduce the least-to-most strategy that prompts LLMs to break down the original problem into a series of sub-problems. Khot et al. [43] further boost this strategy, by assigning each sub-problem to the corresponding LLM that is specifically optimized for it. Finally, few-shot prompting, e.g., Few-shot CoT [44] and Complex CoT [45], has also been studied to enhance the reasoning performance of LLMs. To further improve the few-shot prompting, the prompt retrieval is proposed to automatically select high-quality examples [46, 47], while the prompt compression is explored to include more examples in restricted context by pruning each example [48].

**Rephrasing-based Method**. The second line of existing work aims to generate additional math data, based on which the mathematical reasoning capability of LLMs can be established via supervised fine-tuning. To address the data scarcity issue, current research mainly focuses on rephrasing the problem or the answer. For the answer rephrasing, Magister et al. [49] adopt the PaLM and GPT-3 to generate CoT math data, resulting in improved performance of the T5 model on math reasoning tasks. To mitigate the inclusion of incorrect answers during the supervised fine-tuning, RFT [50] introduces a rejection sampling strategy, whereas AFT [51] trains an LLM to categorize them. Regarding the problem rephrasing, WizardMath [30] proposes a reinforced evol-instruct method. It instructs ChatGPT and trains a new LLM to rephrase the problem, equipped by a reward model for evaluating the quality of generated problems. Combining the rephrasing of problems and answers together, MuggleMATH [31] builds the AugGSM8K dataset based on prompting GPT-3.5 and GPT-4. MetaMath [33] develops a question bootstrapping method based on LLMs, unifying rephrasing, self-verification [52], FOBAR [53], and answer augmentation strategies, obtaining the MetaMathQA. Xwin-Math [54] is a peer study that significantly enhances the reasoning capacity of LLMs using problems generated by GPT-4 Turbo. In contrast, our work focuses on generating verifiable problems through controllable mutations, rather than relying entirely on the GPT model.

Our proposed method also falls into this category. In contrast to existing methods directly prompting LLMs to rephrase the problem, we mutate the problem in the formal symbolic space, resulting in a more controllable mutation mechanism that ensures both the validity and diversity of the generated problems. Moreover, the quality of reasoning paths is also guaranteed by the symbolic solvers.

**Tool-based Method.** Tool-based methods aim to enhance the math solving performance of LLMs by instructing them to use external math tools. For instance, PoT (Program of Thought) [55] and PAL [56] propose to prompt the LLMs to delegate the computation to a program interpreter, which can be executed to obtain the final answer. To further improve the tool-using ability of LLMs, MathCoder [57] constructs a math dataset containing problems and their code-based solutions for the supervised fine-tuning; MAmmoTH [32] builds a dataset that combines CoT and PoT reasoning, enabling LLMs to perform hybrid inference. Since the interaction with math tools can further boost the performance of LLMs, TVA [58] includes the Isabelle theorem prover to check each reasoning step and guide the reflection of LLMs; Tora [59] generates interactive tool-use trajectories on mathematical datasets and then performs imitation learning on the annotations.

Our proposed method shares some similarities with tool-based approaches as both involve symbolic solvers. However, rather than using external tools to solve mathematical problems, our approach aims

to explore the inherent reasoning capability of LLMs. Therefore, symbolic solvers are only used to ensure the validity of the generated data as well as the correctness of the generated reasoning paths.

## 6   Limitations

**The Capability of Symbolic Solvers**. The effectiveness of our approach significantly hinges on the symbolic solvers. However, existing mathematical tools (e.g., Z3 [20], SymPy [23], and SciPy [24]) face limitations when it comes to expressing and solving a wide array of mathematical problems. For instance, the Z3 SMT solver struggles with expressing higher-order concepts like gcd and lcm, while the SymPy encounters difficulties in solving inequalities involving multiple variables. In our framework, we integrate five mathematical tools, i.e., Z3, CVC4 [60], MathSAT [22], SymPy, and SciPy, and employ SMT-LIB [19] as a unified formal language to enhance the performance of symbolic solving.

**The Expressiveness of Mutations**. The mutation operators used within our framework remain limited, especially in generating more difficult problems (e.g., college- and even IMO-level math problems). One of our future work is to introduce more mutation operators, further increasing the problem difficulty. A possible strategy is the problem fusion [61], which fuses two formal problems into a single, new problem, rather than merely modifying an individual problem. Moreover, the informalization facilitated by LLMs can effectively mitigate the unnaturalness issue stemming from brute-force fusion.

**The Dependence on GPT-4**. GPT-4 is involved in our framework to carry out the informalization and generate the reasoning paths. We also consider the possible solutions that the dependence on GPT-4 can be gradually removed. First, by leveraging our generated formal-informal pairs, we can fine-tune a new LLM specifically for the informalization. Second, it is possible to bypass the generation of reasoning paths, through curriculum learning [62, 63] instead of supervised fine-tuning. Particularly, the reward in the curriculum learning can be determined by whether the generated solution is consistent with symbolic solvers , and the curriculum progresses by incorporating problems of various difficulty levels.

## Broader Impact

The paper aims to advance the field of math data generation. There are many potential societal consequences of our work, and we firmly believe that the majority of these impacts are positive and none which we feel must be highlighted here.

## 7   Conclusion

This paper explores the question of whether sufficient exposure to high-quality mathematical data could enhance LLMs' inherent mathematical reasoning capability. We identify a key challenge in balancing diversity and validity in current math problem generation methods. To tackle this challenge, we propose a neuro-symbolic framework that initially generates formal mathematical problems and then informalizes them back into natural language versions. By casting the data generation into the formal language space, the diversity and validity of the generated math problems can be effectively ensured by the systematic sampling and symbolic solvers. Building upon this, we carefully devise a mutation mechanism, establishing the math dataset encompassing various difficulty levels, and prompt the LLMs to accomplish informalization. Through empirical experiments, we demonstrate that our neuro-symbolic data generation framework significantly enhances the performance of various LLMs in mathematical reasoning tasks, surpassing the current state-of-the-art open-source models. The results also suggest a promising pathway for further enhancing LLMs' mathematical capabilities.

In future work, we intend to expand the expressiveness of mutations and enhance the capability of symbolic solvers to support more types of problems, such as inequality problems. Our goal is to offer a data generation framework to automatically generate high-quality, supervised datasets for LLMs. We expect that our neuro-symbolic data generation framework can provide a potential solution for LLMs to solve the problem of data scarcity, and thereby facilitate in building more LLMs in downstream tasks. Further, our framework has the potential to be integrated with recent studies [64], which only require problems and final answers.

## Acknowledgment

We appreciate the anonymous reviewers for their valuable insights and helpful comments. This work is supported by the National Natural Science Foundation of China (Grants #62025202), the Frontier Technologies R&D Program of Jiangsu (BF2024059), and the Key Program of Jiangsu Science Foundation (BK20243012). Xian Zhang (zhxian@microsoft.com) and Xiaoxing Ma (xxm@nju.edu.cn) are the corresponding authors.

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

## A    Broader Impacts

The paper aims to advance the field of math data generation. There are many potential societal consequences of our work, and we firmly believe that the majority of these impacts are positive and none which we feel must be highlighted here.

## B    Examples of Simplification

---

**Example 2: Simplification strategy on Problem (No. 41) of the GSM8K dataset.**

**Autoformalization:**
Translate the natural language problem into SMT-LIB language: "Sara bought a pair of shoes for $50.00 and a dress for $200.00. If Rachel has twice the amount that Sara spent in total, how much is Rachel's budget?"

- - - - - - - - - - - - - - - - - - - - - - - - - - - - - - - - - - - - - - - - - - - - - - - - - - - - - - - - - -

**GPT-4 output:**

```
(declare-fun sara_shoes_cost () Real)
(declare-fun sara_dress_cost () Real)
(declare-fun sara_total_cost () Real)
(declare-fun rachel_budget () Real)
(assert (= sara_shoes_cost 50.0))
(assert (= sara_dress_cost 200.0))
(assert (= sara_total_cost (+ sara_shoes_cost sara_dress_cost)))
(assert (= rachel_budget (* 2 sara_total_cost)))
(check-sat)
(get-value (rachel_budget))
```

- - - - - - - - - - - - - - - - - - - - - - - - - - - - - - - - - - - - - - - - - - - - - - - - - - - - - - - - - -

**Mutation & Informalization:**

```
(declare-fun sara_dress_cost () Real)
(declare-fun sara_shoes_cost () Real)
(declare-fun sara_total_cost () Real)
(assert (= sara_dress_cost 200.0))
(assert (= sara_shoes_cost 50.0))
(assert (= sara_total_cost (+ sara_shoes_cost sara_dress_cost)))
(check-sat)
(get-value (sara_total_cost))
```

Translate the math problem formulated with SMT-LIB back to a natural language problem.

- - - - - - - - - - - - - - - - - - - - - - - - - - - - - - - - - - - - - - - - - - - - - - - - - - - - - - - - - -

**GPT-4 output:**
Sara went shopping and bought a dress for $200.00 and a pair of shoes for $50.00. What is the total amount Sara spent on her shopping trip?

---

We elucidate the simplification strategy through Example 2, which is selected from the GSM8K dataset. In this formal problem, we can conduct one step of Gaussian elimination, i.e., randomly solving and removing a variable (e.g., `rachel_budget` $= 500$). Then, we derive the new problem with its informalized version. Furthermore, we present another example, specifically Example 3, sourced from the MATH dataset. Here, the new problem is generated through the simplification strategy that randomly performs some calculations.

## C    Examples of Informalization

Various prompts yield different styles of informalization. To illustrate this, we provide two specific examples as follows. In Example 4, we do not refresh the variable, and instruct the informalization by adding the prompt "Ensure to be a math word problem." The result of GPT-4's informalization tends

> **Example 3: Simplification strategy on Problem (No. 496) of the MATH dataset.**
>
> **Autoformalization:**
> Translate the natural language problem into SMT-LIB language: "Compute $\$\backslash\backslash$displaystyle $\backslash\backslash$frac{2+4-8+16+32-64}{4+8-16+32+64-128}\$."
>
> - - - - - - - - - - - - - - - - - - - - - - - - - - - - - - - - - - - - - - - - - - - - - - - - - -
>
> **GPT-4 output:**
>
> ```
> (declare-fun num () Real)
> (declare-fun den () Real)
> (declare-fun fraction () Real)
> ; Define the numerator and denominator expressions
> (assert (= num (+ 2 4 (- 8) 16 32 (- 64))))
> (assert (= den (+ 4 8 (- 16) 32 64 (- 128))))
> ; Define the fraction as the numerator divided by the denominator
> (assert (= fraction (/ num den)))
> (check-sat)
> (get-value (fraction))
> ```
>
> - - - - - - - - - - - - - - - - - - - - - - - - - - - - - - - - - - - - - - - - - - - - - - - - - -
>
> **Mutation & Informalization:**
>
> ```
> (declare-fun num () Real)
> (declare-fun den () Real)
> (declare-fun fraction () Real)
> ; Define the numerator and denominator expressions
> (assert (= num (+ (- 2) 16 (-32))))
> (assert (= den (+ (- 4) 32 (- 64))))
> ; Define the fraction as the numerator divided by the denominator
> (assert (= fraction (/ num den)))
> (check-sat)
> (get-value (fraction))
> ```
>
> Translate the math problem formulated with SMT-LIB back to a natural language problem.
>
> - - - - - - - - - - - - - - - - - - - - - - - - - - - - - - - - - - - - - - - - - - - - - - - - - -
>
> **GPT-4 output:** Calculate the value of the fraction $\backslash\backslash(\backslash\backslash\backslash$frac{-2 + 16 - 32}{-4 + 32 - 64}$\backslash\backslash)$.

to envelop the problem with common knowledge, culminating in the creation of a math word problem. In contrast, Example 5 involves refreshing each variable and eliminating the prompt "Ensure it is a math word problem." Hence, its corresponding informalization renders the new problem more abstract and devoid of extraneous elements.

## D  Additional Experimental Details

**Generation Details.** In the GSM8K dataset, each problem can be accurately formalized into the SMT-LIB format and successfully mutated into a new version. However, some problems, particularly in the precalculus and geometry categories of the MATH dataset, cannot be formalized or mutated effectively. Overall, out of a total of 7,500 problems, 822 cannot be formalized into the SMT-LIB format, and approximately 3,600 formalizations are inaccurate although they remain usable for the data generation. To address this issue, we strategically added a proportional number of mutated problems derived by directly prompting GPT-4, bypassing solution verification. The detailed counts of these problems are presented in Table 4.

**Training Details.** In this paper, we fully fine-tune the LLAMA-2-7B and LLAMA-2-13B models using four H800 NVIDIA GPUs. Each model is trained for 3 epochs with a batch size of 128 and a learning rate of 2e-5. For the fine-tuning of the LLAMA-2-70B model, we adopt the QLoRA [65] method with a learning rate of 1e-4. The rank and alpha of LoRA [66] are set to 96 and 16, respectively, with a dropout rate of 0.05 between the two matrices. The LoRA modules are added to both the attention and MLP layers. The 70B model is fine-tuned using eight A800 NVIDIA GPUs.

Example 4: Informalization to be a math word problem (Problem No.14 in GSM8K).

```
(declare-fun pages_per_minute () Real)
(declare-fun total_pages () Int)
(declare-fun time_hours () Int)
(assert (= pages_per_minute (/ 2 5)))
(assert (= total_pages 144))
; (time_hours = (((total_pages / pages_per_minute) * 1/60) * 1/2))
(assert (= time_hours (* (* (/ total_pages pages_per_minute)
                (/ 1 60)) (/ 1 2))))
(check-sat)
(get-value (time_hours))
```

Translate the math problem formulated with SMT-LIB back to a natural language problem. Please ensure to be a math word problem.

- - - - - - - - - - - - - - - - - - - - - - - - - - - - - - - - - - - - - - - - - - - - - - -

**GPT-4 output:**
Jamie has a book with 144 pages that she wants to read. She reads at a pace of 2/5 pages per minute. If she reads for half the time it would normally take her to read the book at this pace, how many hours will she have read?

Example 5: Informalization to be a pure math problem (Problem No.14 in GSM8K).

```
(declare-fun x_0 () Real)
(declare-fun x_1 () Int)
(declare-fun x_2 () Int)
(assert (= x_0 (/ 2 5)))
(assert (= x_1 1))
; (x_2 = (2/5 * (x_1 / x_0)))
(assert (= x_2 (* (/ 2 5) (/ x_1 x_0))))
(check-sat)
(get-value (x_2))
```

Translate the math problem formulated with SMT-LIB back to a natural language problem.

- - - - - - - - - - - - - - - - - - - - - - - - - - - - - - - - - - - - - - - - - - - - - - -

**GPT-4 output:**
A fraction representing a number is 2/5. If you have the number 1 and you want to find out how many times this fraction fits into it, what would be the result?

Below is an instruction that describes a task. Write a response that appropriately completes the request.\n \n ### Instruction:\n{instruction}\n\n### Response:\n

Table 4: Detailed count of problems generated by GPT-4 and mutated during the generation process of the MATH dataset. The numbers 1-7 correspond to Algebra, Counting and Probability, Geometry, Intermediate Algebra, Number Theory, Prealgebra, and Precalculus, respectively.

| Category | 1 | 2 | 3 | 4 | 5 | 6 | 7 | Total |
|---|---|---|---|---|---|---|---|---|
| # LLM generated | 20.4K | 9.2K | 6.4K | 15.4K | 8.4K | 12.4K | 7.8K | 80K |
| # Mutated | 97.4K | 36.2K | 31.2K | 60.3K | 40.7K | 59.2K | 24.1K | 350K |

When fine-tuning the Mistral 7B model, the same training settings as LLAMA-2-7B are used, except

for the learning rate, which is set to 5e-6. Moreover, we adopt the instruction template Prompt 1 used in Alpaca [67] for fine-tuning each model.

> **Prompt 2: Testing Prompt**
>
> Below is an instruction that describes a task. Write a response that appropriately completes the request.\n\n### Instruction:\n{instruction}\n\n### Response:\n

**Evaluation Details**. We evaluate each fine-tuned model using a zero-shot evaluation protocol with the corresponding recommended instruction template. Our model is evaluated using the following instruction template Prompt 2, which is consistent with our training instruction template.

For answer extraction and accuracy calculation, we follow the code of WizardMath [30] to extract the answer after the phrase "The answer is".

**Symbolic Solvers**. We integrate five symbolic solvers, Z3, CVC4, MathSAT, SymPy, and SciPy based on the PySMT framework [68]. To be specific, the PySMT intrinsically includes Z3, CVC4, and MathSAT, and we further extend its support to encompass SMT-LIB version 2.5, which incorporates more commands like `define-rec`. Next, we proceed to serialize the SMT-LIB format into SymPy expressions, and attempt to find solutions using SymPy's `solve` function. In addition, the SymPy expressions are also encoded as NumPy [69] functions, thereby enabling the using of SciPy's optimization modules, such as `differential_evolution` and `minimize`. Note that we introduce a fuzzy-logic-like strategy [70–72] in the encoding, which combines the equalities and inequalities into a loss function, subsequently enabling optimization methods for problem-solving tasks.

# E    Additional Experimental Results

## E.1    Detailed results on MATH dataset

We present detailed results across different categories in the MATH dataset in Table 5. The numbers 1-7 correspond to Algebra, Counting and Probability, Geometry, Intermediate Algebra, Number Theory, Prealgebra, and Precalculus, respectively. The results indicate that Algebra is easier to improve, as evidenced by its higher mutation success rate. It is worth noting that improvements in Counting and Number Theory are reasonable because related mutation operators (e.g., binomial, gcd, lcm, etc.) are included in our framework. However, Precalculus and Geometry are still not well-supported. For example, the concept of triangle cannot currently be correctly expressed in the SMT-LIB format, resulting in relatively low improvement rates.

Table 5: Comparison of performance between MetaMath and our method across different categories of MATH dataset. The used base model is Mistral-7B. The best performance is in bold.

| Category | 1 | 2 | 3 | 4 | 5 | 6 | 7 |
|----------|------|------|------|------|------|------|------|
| MetaMath | 41.4 | 23.6 | 15.4 | 20.7 | 47.9 | 15.3 | 22.5 |
| Ours | **59.6** | **35.4** | **16.4** | **32.8** | **58.5** | **18.5** | **25.0** |
| Δ | +18.2 | +11.8 | +1.0 | +12.1 | +10.6 | +3.2 | +2.5 |

## E.2    Comparison to tool-based methods

We compare our model with the tool-based models in Table 6. Although tool-based models achieve good performance with tools, they meet severe performance degradation when tools are not available. This result indicates that training a model using code-based and language-based rationales does not necessarily enhance the intrinsic reasoning ability; instead, it often promotes excessive dependence on external tools. For datasets that involve complex calculations, such as the MATH dataset, tool-based methods offer certain advantages due to their utilization of the strong capabilities of external tools. For datasets that emphasize knowledge reasoning but involve simpler calculations, such as the

Table 6: Performance comparison among tool-based methods and our methods. We report the performance of tool-based methods w/ and w/o tools. The best performance is in bold.

| Model | Base Model | GSM8k | MATH |
|---|---|---|---|
| Tora | LLaMA2-7B | 68.8 | 40.1 |
| MAmmoTH | LLaMA2-7B | 53.6 | 31.5 |
| MAmmoTH(w/o tools) | LLaMA2-7B | 50.5 | 10.4 |
| Tora | CodeLLaMA-7B | 72.6 | **44.6** |
| MAmmoTH | CodeLLaMA-7B | 59.4 | 33.4 |
| MAmmoTH(w/o tools) | CodeLLaMA-7B | 22.1 | 7.6 |
| MAmmoTH | Mistral-7B | 75.0 | 40.0 |
| MAmmoTH(w/o tools) | Mistral-7B | 61.9 | 17.5 |
| Ours | LLaMA2-7B | 79.2 | 28.8 |
| Ours | Mistral-7B | **86.8** | 37.3 |

GSM8K dataset, they underperform our proposed methods, proving that their reasoning ability is still inadequate.

### E.3 Experiments on DyVal Datasets

We also compare our models and existing mathematical reasoning models using DyVal [9] datasets to further evaluate the generalization ability of our models. DyVal is a flexible evaluation protocol for dynamic evaluation of LLMs, which generates evaluation samples with controllable complexities using directed acyclic graphs (DAGs). We focus on Arithmetic tasks using three DAGs' orders: topological (TOPO), reversed topological (REVERSED), and random orders (RAND). Following the same setting, we generate testing four increased complexity levels {D1, D2, D3, D3}, with tree depths and widths set to (2, 2), (3, 2), (3, 3), (4, 2). The performance comparison between models fine-tuned on LLaMA2-7B and Mistral-7B are respectively shown in Table 7 and Table 8. The results show that our models achieve the best performance in 11/12 cases and give a competitive performance in the remaining one case. As the complexity increases, our models achieves a relatively robust performance compared with the other models.

Table 7: Performance comparison of models fine-tuned on LLaMA2-7B using DyVal datasets with four complexity levels and three graph orders. The best performance is in bold.

| Model | Arithmetic-Topo | | | | Arithmetic-Reversed | | | | Arithmetic-Rand | | | |
|---|---|---|---|---|---|---|---|---|---|---|---|---|
| | D1 | D2 | D3 | D4 | D1 | D2 | D3 | D4 | D1 | D2 | D3 | D4 |
| WizardMath | 74.0 | 55.4 | 21.4 | 19.9 | 71.0 | 50.6 | 23.2 | 7.8 | 74.2 | 49.0 | 19.6 | 10.7 |
| MAmmoTH | 77.3 | 35.1 | 13.7 | 8.1 | 81.8 | 27.7 | 13.1 | 4.8 | 78.3 | 26.4 | 12.7 | 3.8 |
| MetaMath | 91.9 | 48.0 | 20.3 | 16.9 | 90.5 | 50.5 | 18.1 | 5.8 | 90.2 | 41.0 | 17.9 | 7.4 |
| Ours | **99.7** | **85.7** | **66.8** | **62.5** | **99.1** | **75.8** | **39.8** | **18.1** | **98.8** | **71.4** | **41.2** | **19.0** |

Table 8: Performance comparison of models fine-tuned on Mistral-7B using DyVal datasets with four complexity levels and three graph orders. The best performance is in bold.

| Model | Arithmetic-Topo | | | | Arithmetic-Reversed | | | | Arithmetic-Rand | | | |
|---|---|---|---|---|---|---|---|---|---|---|---|---|
| | D1 | D2 | D3 | D4 | D1 | D2 | D3 | D4 | D1 | D2 | D3 | D4 |
| MAmmoTH | 90.2 | 66.7 | 29.9 | 32.8 | 92.2 | 64.8 | 25.3 | 18.8 | 91.0 | 62.2 | 23.9 | 18.2 |
| MetaMath | 97.8 | 75.7 | 43.0 | 45.6 | **99.5** | 78.6 | 45.6 | 37.2 | 98.3 | 76.3 | 44.6 | 38.8 |
| Ours | **99.6** | **91.3** | **74.0** | **71.5** | 99.3 | **85.4** | **56.8** | **49.2** | **98.4** | **87.5** | **68.6** | **47.2** |

### E.4 Diversity Gain across Various Difficulty Levels

To illustrate the need for various difficulty levels, we calculate the diversity gain relative to the original dataset for each difficulty level. We apply the same method as in MetaMath [33] to compute diversity

gain but use the BERT model [73] as a feature extractor instead. We select data budgets of 35K, 50K, and 100K, respectively, and investigate the diversity across four levels, from level-1 to level-4. Additionally, we create a mixed version by sampling data from all levels. The results are shown in Figure 5, and we can observe that: (1) The higher the difficulty level, the greater the diversity of generated data; (2) The mixed version increases with the growing data budget, and achieves the highest diversity gain.

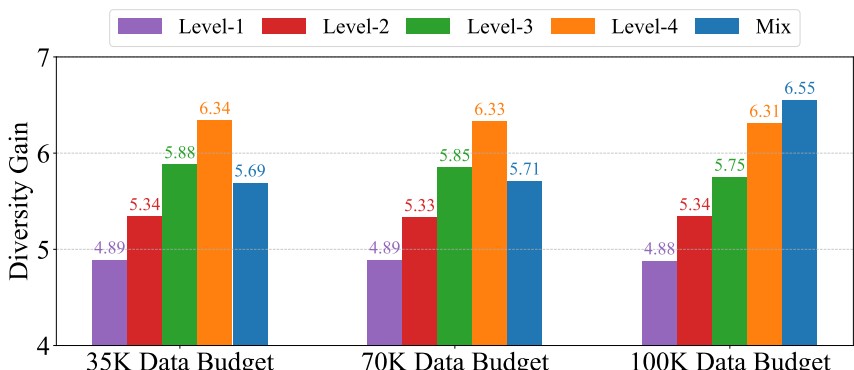

Figure 5: The diversity gain across all difficulty levels. The results indicate that the diversity gain of the Mix version continues to increase and reaches the highest compared with alternatives as the data budget increases.

