# OpenReview forum: "Neuro-Symbolic Data Generation for Math Reasoning"
_NeurIPS.cc/2024/Conference — NeurIPS 2024 poster_

### Official Review · Reviewer_UMpg · 2024-06-29

**Soundness:** 3
**Presentation:** 3
**Contribution:** 3
**Rating:** 7
**Confidence:** 3

**Summary:**

This paper introduces a methodology for generating mathematics data in a neurosymbolic fashion. Starting with existing math problems, they perform two different mutation operations: simplification and complication. The simplification operation performs variable and expression unfolding, whereas the complication operation adds additional complexity to the problem statements by adding auxiliary variables. Both operations are performed on the formalized problem statement (which is formalized in SMT-lib) format.  Symbolic solvers attempt to solve the formal problem in a way which mimicks GPT-4 generated solution. The resulting procedure is applied with seed mathematics problems (like from GSM8K, MATH) and produces a dataset which they demonstrate to be of higher quality than prior approaches when fine-tuning open source models for mathematical reasoning.

**Strengths:**

1. Performing the data synthesis process with formal specifications and using symbolic solvers is a nice idea, and is useful towards generating higher quality training datasets.

2. The evaluations indicate that their data generation process produces higher quality data compared to other existing approaches for synthetic data generation. This in particular indicates that the work is fairly significant and resultant dataset is likely to be used among the research community.

3. In general, the writing is clear and easy to follow. This makes the paper easily digestable. I noted some typos, see in a later section.

**Weaknesses:**

1. In Section 3 you mention that a faithful conversion from the natural language to the formal specification cannot be automatically checked, and mention that you use GPT-4 to generate an informal solution and measure this against the solution produced by the symbolic solver. I am skeptical of a zero positive rate, and there does not seem to be any numerical results indicating a zero false positive rate. Is the consistency check performed by checking if the numerical solutions are the same? Or if the reasoning path produced by GPT-4 and by the solver are similar? If the latter case, how can you perform this check?

2. In the abstract the authors mention that a main component of their approach is the projected MCMC method. However, it is only mentioned briefly in the main content of the paper (lines 100-104) and the definition of what projected MCMC is is not included. I think it would be beneficial to include a more detailed discussion of this in the paper.

3. While grounding the methodology with formal specifications in SMT-Lib is a nice idea, they mention that for the MATH dataset the symbolic solvers are not usually capable of solving such problems, which hampers the applicability of this method. For the MATH dataset, they mention that of 7500 problems, 822 cannot be formalized in the SMT-Lib format and ~3600 are inaccurate so they bypass the solution verification (a central component of their approach) by prompting GPT-4 directly. Thus, I am worried about the scalability of this approach to harder mathematics (or reasoning) problems.

Typos:
Line 69: involes -> involves
Line 157: effecitveness -> effectiveness
Table 3 caption: illusrate -> illustrate
Line 338: repsectively -> respectively

**Questions:**

1. What do GPT-4 generated solutions look like? No examples are provided in the paper. Similarly, no solution outputs from the symbolic solvers are provided. Figure 1 mentions that reasoning paths are produced by GPT-4 but are never shown.

2. In the complication section, can you provide specific details about the choice of interpreted functions $foo$? It is only mentioned at the top of page 4 as far as I know. It would be good to include (perhaps in the appendix).  Why is a random (uniform) selection among them a good choice? It might be unnatural to include too many problems including arcsin, for example.

3. Which symbolic solvers are used? SMT Solvers may not return a reasoning path for solution to a problem, only indicating a truth value and a final answer. On page 2 they mention that sympy & scipy are not directly amenable for smt-lib but it can be extended pretty easily. Are sympy and scipy used?

**Limitations:**

I believe the authors have adequately addressed limitations of their work.

---

> ### Author Rebuttal · Authors · 2024-08-07
>
> **Dear Reviewer XHjb:**
>
> Thank you for the valuable feedback on our paper. We appreciate the time and effort you have put into reviewing our work and we are grateful for encouraging comments such as nice idea, significant work, and clear writing. We have carefully read your review and addressed your concerns as follows.
>
> **[Weakness #1] False positive rate of autoformalization**
>
> We apologize for the confusion about the false positive rate. The consistency check is performed by directly checking the numerical solutions. A false positive refers to that the formal problem is inconsistent with its informalized version while their derived solutions (by symbolic solver and GPT-4, respectively) are consistent. Given the rarity of such instances, a (nearly) zero false positive rate is reasonable. We will further illustrate the definition of the false positive in the revision.
>
> **[Weakness #2] More details about projected MCMC**
>
> Thanks for the comments. We will include a more detailed description of projected MCMC in the revision.
>
> **[Weakness #3] Scalability of the approach**
>
> Sorry for the unclear statements. In the MATH dataset, 822 problems are fully unsolvable by the symbolic solver, hence we bypass solution verification for these cases. As to ~3600 problems with inaccurate formalizations, although their formal versions are inconsistent to original versions, they can still serve as seed problems for subsequent mutation, and solution verification is still conducted. The detailed data sizes for purely LLM-generated and mutated problems, along with an ablation study of removing purely LLM-generated data, are presented in Appendix E (Table 4) and Appendix F (Table 5). We will further clarify this in the revision.
>
> **[Question #1] Details of generated solutions**
>
> Thanks for the comment. We use the standard setting for the solution generations (CoT prompt, temperature=0.7, top_p=0.95). An example of formalized problem, symbolic solver's solution, informalized problem, and GPT-4's reasoning path is shown in the following.
> ```
> # Formalized Problem
> (declare-fun a () Real)
> (declare-fun b () Real)
> (declare-fun f (Real) Real)
>
> ; Define the piecewise function f(x)
> (define-fun f ((x Real)) Real
>   (ite (> x 2) (+ (* a x) 3)
>     (ite (< x (- 2)) (- (* 2 x) b)
>       (- x 5))))
>
> ; Assert continuity at x = 2
> (assert (= (f 2) (f 2.0000001)))
>
> ; Assert continuity at x = -2
> (assert (= (f (- 2)) (f (- 2.0000001))))
>
> ; Find a + b
> (declare-fun sum_ab () Real)
> (assert (= sum_ab (+ a b)))
>
> ; Solve the constraints
> (check-sat)
> (get-value (a b sum_ab))
>
> # Symbolic solver
> (-3.0, 3.0, 0.0)
>
> # Informalized problem
> Let \\[f(x) = \\left\\{\n\\begin{array}{cl}\nax + 3, &\\text{ if } x > 2, \\\\\nx - 5, &\\text{ if } -2 \\leq x \\leq 2, \\\\\n2x - b, &\\text{ if } x < -2.\n\\end{array}\n\\right.\\]\n\nDetermine the values of $a$ and $b$ for which the function $f(x)$ is continuous at $x = 2$ and $x = -2$, and then calculate the sum of $a$ and $b$.
>
> # GPT-4 reasoning paths
> For the function $f(x)$ to be continuous at $x = 2$, the values of the function as it approaches 2 from the left and the right must be the same. Thus the value of the function for $x > 2$ must equal the value of the function for $-2 \\leq x \\leq 2$ when $x = 2$.\n\nSo we set $ax + 3$ equal to $x - 5$ when $x = 2$:\n\n$2a + 3 = 2 - 5$\n\n$2a = -6$\n\n$a = -3$.\n\nSimilarly, for $f(x)$ to be continuous at $x = -2$, the values of the function for $-2 \\leq x \\leq 2$ must equal the value of the function for $x < -2$ when $x = -2$.\n\nSo we set $x - 5$ equal to $2x - b$ when $x = -2$:\n\n$-2 - 5 = 2(-2) - b$\n\n$-7 = -4 - b$\n\n$b = 3$.\n\nNow that we have the values of $a$ and $b$, we can find their sum:\n\n$a + b = -3 + 3 = 0$.\n\nThe answer is 0.
> ```
>
> **[Question #2]  Details about the choice of interpreted functions**
>
> We apologize for the lack of details. In our implementation, we first identify which functions are used in the problem and then determine the interpreted functions accordingly. For instance, if a problem involves sine functions, the interpreted trigonometric functions will be randomly (and uniformly) selected, while other irrelevant functions (e.g., lcm, gcd) will be excluded. This step is adapted from the theory/logic selection in PySMT (see [the code piece](https://github.com/pysmt/pysmt/blob/master/pysmt/operators.py)).
>
> **[Question #3] Details about symbolic solvers**
>
> Yes, SciPy and SymPy are used in our framework. In our implementation, we first serialize the SMT-LIB format into SymPy expression, and then use SymPy `solve` and SciPy `minimize` function to solve the problem. More implementation details are shown in Appendix E.

---

> > ### Comment · Reviewer_UMpg · 2024-08-08
> >
> > I thank the authors for their response which has addressed most of my questions.
> >
> > >  In the MATH dataset, 822 problems are fully unsolvable by the symbolic solver, hence we bypass solution verification for these cases. As to ~3600 problems with inaccurate formalizations, although their formal versions are inconsistent to original versions, they can still serve as seed problems for subsequent mutation, and solution verification is still conducted.
> >
> > By this, you mean that the symbolic solver does not terminate in producing a numerical answer, right? And as for the 3600 problems whose formalizations are judged to be inaccurate, this is because the numerical answer computed by the solver does not match the ground truth answer in MATH? Can you share numbers as to how many problems from each level 1-5 in MATH fall under the 822 or 3600?

---

> > > ### Author Response · Authors · 2024-08-09
> > > **Response to Reviewer UMpg**
> > >
> > > > You mean that the symbolic solver does not terminate in producing a numerical answer, right?
> > > >
> > >
> > > Yes. The formalized problem is identified as unsolvable if the symbolic solver does not terminate in producing a valid answer (it may produce a symbolic answer).
> > >
> > > > As for the 3600 problems whose formalizations are judged to be inaccurate, this is because the numerical answer computed by the solver does not match the ground truth answer in MATH?
> > > >
> > >
> > > Yes.
> > >
> > > > Can you share numbers as to how many problems from each level 1-5 in MATH fall under the 822 or 3600?
> > > >
> > >
> > > Yes. The detailed results are shown in the following.
> > >
> > > | Level 1 | Level 2 | Level 3 | Level 4 | Level 5 | Total |
> > > | --- | --- | --- | --- | --- | --- |
> > > | 45 | 46 | 115 | 137 | 479 | 822 |
> > > | 198 | 574 | 746 | 869 | 1250 | 3637 |

---

> > > > ### Comment · Reviewer_UMpg · 2024-08-09
> > > >
> > > > Thanks for answering these questions. I think the authors have addressed all the questions I had in a satisfactory way. Consequently I have increased my score.

---

### Official Review · Reviewer_upsU · 2024-07-12

**Soundness:** 4
**Presentation:** 4
**Contribution:** 4
**Rating:** 7
**Confidence:** 5

**Summary:**

This paper proposes a neural-symbolic framework to generate valid and diverse mathematical training data at scale. The framework consists of three steps: formalization, mutation, and reformalization. The first two steps are achieved using symbolic solvers, while the last step is accomplished using large language models (LLMs). The mutation step is controllable. The experiments demonstrate that the proposed method is more effective, efficient, and scalable.

**Strengths:**

- The paper is clearly structured and easy to follow.

- The proposed method is novel, particularly the reliable and controllable mutation mechanism. I believe this will be very beneficial to the community and has the potential to scale up significantly beyond the current capabilities.

- The experiments are solid and contain interesting findings, such as the increasing lengths of the generated solutions and the accuracy improvement with the increasing difficulty of the problems.

**Weaknesses:**

- It would be beneficial to compare the proposed method with other mathematical models that have comparable SFT data sizes, such as those in [1][2].

 [1] MathScale: Scaling Instruction Tuning for Mathematical Reasoning

 [2] Augmenting Math Word Problems via Iterative Question Composing

**Questions:**

What is the cost of synthesizing 860k data? For example, what is the CPU cost (e.g., how many CPU hours) needed for the symbolic solvers?

**Limitations:**

Yes

---

> ### Author Rebuttal · Authors · 2024-08-07
>
> **Dear Reviewer upsU:**
>
> Thank you for the valuable feedback on our paper. We appreciate the time and effort you have put into reviewing our work and we are grateful for encouraging comments such as the clear paper structure, novel method, solid experiments, and interesting findings. We have carefully read your review and addressed your concerns as follows.
>
> **[Weakness #1] Compared with other models that have comparable SFT data sizes**
>
> We reproduce the MathScale and MMIQC based on the repositories [1] and [2] based on the Mistral base model. Note that both MathScale and MMIQC methods generate approximately 2M data to fine-tune the Mistral 7B model, while our proposed method only generates 860K data.
>
> The results show that our proposed data generation method achieves higher performance with less volume of generated data, demonstrating the efficiency and effectiveness of our proposal. We will add these two comparison methods to our final paper.
>
> | Model | #Dataset | GSM8K | MATH |
> | --- | --- | --- | --- |
> | MathScale | ~1,999K | 74.0 | 34.5 |
> | MMIQC | ~2,294K | 60.5 | 36.0 |
> | Ours | 860K | 86.8 | 37.3 |
>
> [1] https://huggingface.co/fdqerq22ds/MathScale-Mistral
>
> [2] https://huggingface.co/datasets/Vivacem/MMIQC
>
> **[Question #1] Cost of  data synthesis**
>
> The cost of data synthesis mainly depends on the calling of GPT-4 (autoformalization, informalization, and reasoning path generation), it requires ~1K tokens cost for each problem generation. As to the CPU cost, in our experiments, we use a 48-core CPU, and take about 770s to mutate 5000 problems from MATH dataset , i.e., 0.154s per each problem. Hence, this part of the consumption is totally acceptable.

---

> > ### Comment · Reviewer_upsU · 2024-08-13
> >
> > Thank you for your detailed responses, which have addressed most of my concerns. I believe I provided a fair rating and intend to maintain it.

---

### Official Review · Reviewer_bEBk · 2024-07-20

**Soundness:** 3
**Presentation:** 3
**Contribution:** 3
**Rating:** 5
**Confidence:** 4

**Summary:**

To solve the dilemma of diversity and validity involved in current math problem generation methods, this paper proposes a neuro-symbolic framework that initially generates formal mathematical problems and then informalizes them back into natural language versions. By casting the data generation into the formal language space, the diversity and validity of the generated math problems can be effectively ensured. Then, a mutation mechanism is adopted to establish the math dataset encompassing various difficulty levels, and prompt the LLMs to accomplish informalization with consistency between formal language problem and its natural language version guaranteed by symbolic solvers. Experimental results demonstrate that the synthetic data can significantly enhance the performance of various LLMs in mathematical reasoning tasks.

**Strengths:**

1.the proposed framework is promising to both guarantee the diversity and accuracy of the synthetic mathematical data, based on the formal language.

2.experimental results have shown the effectiveness of the whole approach.

**Weaknesses:**

1.the motivation of this work is obvious, but it is not clear how to convert the natural language into the well-formulated SMT-LIB language. The cost and examples about the whole pipeline should be depicted in the paper. Also, I am wandering if the formulation process is not also compatible for all the natural language questions, should human effort be involved for controlling the quality?

2.the whole approach is also similar to existing work that using program with tools to process natural language questions. It is not clear what is the major difference between the SMT-LIB language and the program-aided language in existing work. Why not using programming language for the whole pipeline?

3.the experimental setting is not fully fair. For the main results, all the compared baselines are trained using 1/2 fewer data. I suggest authors to report the results using the same data amount as the best-performed baselines for comparison.

**Questions:**

Please refer to the weaknesses

---

> ### Author Rebuttal · Authors · 2024-08-07
>
> **Dear Reviewer bEBk:**
>
> Thank you for the valuable feedback on our paper. We appreciate the time and effort you have put into reviewing our work, and we are grateful for encouraging comments such as promising framework and effective approach. We have carefully read your review and addressed your concerns as follows.
>
> **[Weakness #1] Examples and costs of the whole pipeline**
>
> For the conversion from natural language to SMT-LIB language, we have included an example of autoformalization, mutation, and informalization in Appendix E (Example 2).
>
> The cost of the whole pipeline primarily depends on the calling of GPT-4 (it needs ~1K tokens cost per problem generated). This cost of calling GPT-4 is similar to the compared data generation methods (e.g., MetaMath). Regarding CPU cost, in our experiments, we use a 48-core CPU and take about 770s to mutate 5000 problems from the MATH dataset, i.e., 0.154s per problem. Therefore, this part of the cost is totally acceptable.
>
> The human effort required during the formalization stage is minimal, involving manually defining a few uninterpreted functions (e.g., arcsin, arccos, gcd, lcm, and etc.) in advance. Apart from this, our entire data generation pipeline is fully automated.
>
> **[Weakness #2] Compared with tool-used methods**
>
> We would like to clarify that our method is fundamentally different from tool-used methods.
>
> Our framework uses formal language to formalize the problem, which ensures the problem is well-structured and thus facilitates the mutation. Benefiting from the formalized problem, we can efficiently mutate and verify problems using symbolic solvers (e.g., SymPy) to ensure the diversity and validity of the newly generated problems.
>
> However, PAL-like and PoT-like methods prompt the LLM to generate a program for solving the problem. Therefore, these methods are not amenable to mutation with solely programs rather than formalized problems.
>
> **[Weakness #3] Comparison on the same data amount**
>
> We would like to first clarify that the generated data from different methods may have different convergence rates (i.e., accuracy curves) as the data size grows, which is also confirmed in Appendix E.2 of MetaMath [1]. Therefore, in Table 1, we directly compare different models according to their released datasets and models because we believe each method has already achieved its (near-)best performance.
>
> For a comparison with equal data size, we have conducted a comparison between our method and MetaMath with the same data budget as an ablation study in Table 3. The result illustrates that our method is still better.
>
> In addition, we will include the comparison with other models that have significantly large SFT data sizes in the revision, i.e., MathScale and MMIQC [2, 3]. Both MathScale and MMIQC methods generate approximately 2M data to fine-tune the Mistral 7B model. The results below show that our proposed data generation method achieves higher performance with less volume of generated data, demonstrating the efficiency and effectiveness of our proposal.
>
> | Model | #Dataset | GSM8K | MATH |
> | --- | --- | --- | --- |
> | MathScale | ~1,999K | 74.0 | 34.5 |
> | MMIQC | ~2,294K | 60.5 | 36.0 |
> | Ours | 860K | 86.8 | 37.3 |
>
> [1] MetaMath: Bootstrap Your Own Mathematical Questions for Large Language Models. ICLR 2024.
>
> [2] MathScale: Scaling Instruction Tuning for Mathematical Reasoning. ICML 2024.
>
> [3] Augmenting Math Word Problems via Iterative Question Composing. ICLR Workshop 2024.

---

> > ### Comment · Reviewer_bEBk · 2024-08-13
> >
> > I have read the response. Thanks for your clarification. I have increased my score.

---

### Official Review · Reviewer_BS8K · 2024-07-20

**Soundness:** 3
**Presentation:** 3
**Contribution:** 3
**Rating:** 7
**Confidence:** 3

**Summary:**

This paper describes a framework to transform natural language math problems into a formal setting (e.g., in SMT-LIB format), mutate the problems in a user-specified way, and auto-informalise those mutated problems into natural language ones. Through this pipeline, a larger synthetic dataset can be generated to bootstrap the math reasoning performance of existing LLMs (e.g., via fine-tuning). Improved performance has been achieved against previous SOTA including MetaMath.

**Strengths:**

- relatively well-written, good performance against prior works
- I especially appreciate the ideo of mutating the problem in a controllable ways, which leads to slightly more transparent LLMs.
- More importantly, a promising scalability of this method has been demonstrated in Figure 4: as the size of the synthesised dataset grows, the performance of the fine-tuned model LLaMA-2-7B is more consistently enhanced (in comparison to the baseline approach MetaMath).

**Weaknesses:**

- As a trade-off for controllability, the math problems need to be formalized into a user-specified structure. This requirement may significantly hinder the proposed approach from being adapted to problems beyond high school competition level.
- Although the authors demonstrate that this approach performs comparably or even better than many existing tool-use methods, I am uncertain if it serves as a suitable replacement for these frameworks, which may be more efficient and reliable. Perhaps we could conduct experiments to explore combining this approach with existing tool-use frameworks?
- The complication process could be better formulated. In a formal setting, I would appreciate a more rigorous definition of validity and diversity, along with a clearer explanation of what has been preserved and enhanced during the mutation process. Additionally, I would like to see more details on how the projected MCMC method enhances the diversity of mutated problems. Including some pseudocode would be helpful.

Minor:
- line 157, typo: 'effecitveness' -> 'effectiveness'

**Questions:**

- As a methodology that heavily relies on fine-tuning that sometimes cause catastrophic forgetting, I am curious to know the performance of the fine-tuned models on other non-math tasks.

**Limitations:**

N.A.

---

> ### Author Rebuttal · Authors · 2024-08-07
>
> **Dear Reviewer BS8K:**
>
> Thank you for the insightful feedback on our paper.  We appreciate the time and effort you have put into reviewing our work, and we are grateful for encouraging comments such as good writing, good performance, and promising scalability. We have carefully read your review and addressed your concerns as follows.
>
> **[Weakness #1]  Limitations of the user-specified structure**
>
> We agree that adapting our framework to convert more general mathematical problems into the SMT-LIB format is challenging. One potential solution is integrating our framework with existing theorem provers like Isabelle or Lean. Specifically, we could use Isabelle or Lean to formalize the problem and then employ tactics such as `by smt` in Isabelle or `lean-smt` in Lean to automatically obtain its SMT-LIB version. This could be a future direction to explore.
>
> **[Weakness #2] Combining the method with tool-use framework**
>
> We appreciate the reviewer’s suggestion, and combining our work with tool-use methods is a promising future direction. Specifically, existing work, such as MAmmoTH [1], has demonstrated that combining the PoT and CoT data can further improve the math problem-solving capability of LLMs. Our initial experiments on AIMO Kaggle competition also confirm it (we achieved top-20 by generating CoT and ToRA [2] data using our approach). We will present a comprehensive analysis of these results in our revision.
>
> [1] MAmmoTH: Building Math Generalist Models through Hybrid Instruction Tuning. ICLR 2024.
>
> [2] ToRA: A Tool-Integrated Reasoning Agent for Mathematical Problem Solving. ICLR 2024.
>
> **[Weakness #3] Definitions of validity and diversity**
>
> Thanks for the comment. Given a formal problem, the validity refers to whether the problem is solvable, and the diversity refers to whether the newly generated problem is semantically different to existing problems. We will add more formal explanations for them in the revision.
>
> To see how the projected MCMC preserves the validity and enhances the diversity, let us start with the M3 problem,  which requires determining auxiliary variables (i.e., $z_1$ and $z_2$) to generate a valid problem.
>
> $$
> (\text{M}_3)\\left\\{
> \\begin{array}{l}
> a(b+c) + z_1 = 152 \\\\
> b(c+a) - z_2 = 162 \\\\
> c(a+b) = 170 \\\\
> a, b, c \in \mathbb{N}^+
> \\end{array}
> \\right.
> $$
>
> If we use the symbolic solver directly, it tends to produce the same result ($a=84, b=1, c=2, z_1 = -100, z_2 = -76$) due to inherent solving biases, causing the newly generated problem duplicated. Projected MCMC first randomly perturbs a variable (e.g., $a=84$ to $a=100$), and then solves the rest variables with the symbolic solver ($a=100, b=70, c=1$, $z_1=-6948, z_2=6908$), deriving a new and different problem. We will include the pseudo-code in our revised paper.
>
> **[Question #1] Results of catastrophic forgetting**
>
> Thanks for pointing this out. We evaluate our model and MetaMath model (both fine-tuned on LLaMA-2 7B) on TruthfulQA (zero-shot),  MMLU (zero-shot),  BBH (average score), IFEval (prompt-level acc.), and HumanEval (pass@1). The results are shown as follows and the result of LLaMA-2 7B is also included as a reference. It can be observed that catastrophic forgetting does occur for all tasks except TruthfulQA.  Especially, the performance of code generation is diminished due to the CoT instruction fine-tuning.
> | Model | TruthfulQA ( MC1 ) | MMLU | BBH | IFEval | HumenEval |
> | --- | --- | --- | --- | --- | --- |
> | LLaMA-2 7B | 24.6 | 40.9 | 39.3 | 20.5 | 13.7 |
> | MetaMath | 27.6 | 32.9 | 36.6 | 14.0 | 0.0 |
> | Ours | 27.0 | 30.4 | 36.5 | 15.5 | 0.0 |

---

> > ### Comment · Reviewer_BS8K · 2024-08-12
> >
> > Many thanks for the elaborated response. All my concerns have been resolved. I will increase my score.

---

### Author Rebuttal · Authors · 2024-08-07

We thank all the reviewers for their in-depth comments, which urge us improving our paper. We will revise the paper accordingly. Here, we summarize our responses to the major issues raised by the reviewers.

Reviewers **BS8K**, **bEBk**, and **XHjb** request further discussion about tool-use framework. First, our  method is fundamentally different from tool-based methods as we aim to formalize the problems allowing for mutation and verification, which tool-based methods cannot do (shown in response to reviewer **bEBk**). Moreover, combining our method with tool-use framework may further improve the capability of LLMs (shown in response to reviewer **BS8K**). We will present a comprehensive analysis of these results in our revision.

Reviewer **bEBk** and **upsU** question the cost of our method. First, our method is automated, only requiring minimal human effort to define a few uninterpreted functions beforehand. Next, the cost of our method mainly depends on the calling of GPT-4 (~1K tokens cost per problem) and CPU cost of mutation (0.154s per problem on 48-core CPU). Therefore, the cost of our method is totally acceptable and similar to the comparison data generation method.

Reviewer **BS8K**, **bEBk**, and **upsU** concern about the experiments in our paper. We first would like to clarify that specific data generation methods can determine the final performance. Therefore, we directly compare different methods with their recommended data scales. To further demonstrate the effectiveness of our method, we add two comparison models trained with millions of training samples, i.e., MathScale [1] and MMIQC [2] (shown in response to reviewer **bEBk** and **upsU**).

Reviewer **BS8K and XHjb** question the details of validity and diversity, autoformalization, projected MCMC, and so on. We have presented the detailed explanations in the corresponding responses. All the details will be further clarified in the revision.

[1] MathScale: Scaling Instruction Tuning for Mathematical Reasoning. ICML 2024.

[2] Augmenting Math Word Problems via Iterative Question Composing. ICLR Workshop 2024.

---

### Decision · Program_Chairs · 2024-09-25

**Decision:**

Accept (poster)

**Comment:**

This paper addresses the important problem of generating informative datasets to train LLMs to reason. In particular, the authors devise a neuro-symbolic pipeline that through auto-formalization generates mathematical problems in a domain-specific language, then "mutates" them and generates paraphrases in natural language. This pipeline is shown to generate data that empirically boosts the performance of a range of LLMs.

Reviewers appreciated the neuro-symbolic direction of the paper, and the provided boost in performance. At the same time, they highlighted a number of potential issues, namely, lack of discussion on scaling with data size and how costly the method could be for users. Furthermore, they requested more formal definitions for concepts of validity and diversity as used in the paper, and details for reproducibility. The authors successfully addressed these concerns during rebuttal, convincing reviewers to increase their scores. While some perplexity still remains around the quality of the presentation of the work and some related works (which however are different in practice as they use tools), I believe that the work brings enough valid contributions to the table to be accepted.